# Revisiting the Optimality of Word Lengths

**Tiago Pimentel**[1,2]    **Clara Meister**[2]    **Ethan Gotlieb Wilcox**[2]
**Kyle Mahowald**[3]    **Ryan Cotterell**[2]
[1]University of Cambridge    [2]ETH Zürich    [3]University of Texas, Austin
{`tiago.pimentel`, `clara.meister`, `ethan.wilcox`, `ryan.cotterell`}@inf.ethz.ch
`mahowald@utexas.edu`

## Abstract

Zipf (1935) posited that wordforms are optimized to minimize utterances' communicative costs. Under the assumption that cost is given by an utterance's length, he supported this claim by showing that words' lengths are inversely correlated with their frequencies. Communicative cost, however, can be operationalized in different ways. Piantadosi et al. (2011) claim that cost should be measured as the distance between an utterance's information rate and channel capacity, which we dub the channel capacity hypothesis (CCH) here. Following this logic, they then proposed that a word's length should be proportional to the expected value of its surprisal (negative log-probability in context). In this work, we show that Piantadosi et al.'s derivation does not minimize CCH's cost, but rather a lower bound, which we term CCH↓. We propose a novel derivation, suggesting an improved way to minimize CCH's cost. Under this method, we find that a language's word lengths should instead be proportional to the surprisal's expectation *plus its variance-to-mean ratio*. Experimentally, we compare these three communicative cost functions: Zipf's, CCH↓, and CCH. Across 13 languages and several experimental settings, we find that length is better predicted by frequency than either of the other hypotheses. In fact, when surprisal's expectation, or expectation plus variance-to-mean ratio, is estimated using better language models, it leads to worse word length predictions. We take these results as evidence that Zipf's longstanding hypothesis holds.

## 1 Introduction

Zipf proposed the idea that languages are optimized to minimize their expected utterance length (Zipf, 1935).[1] Under this hypothesis, a word's length should be inversely proportional to its **frequency**. Indeed, this relationship has been attested across a wide variety of the world's languages (Grzybek, 2015; Bentz and Ferrer-i-Cancho, 2016, *inter alia*).

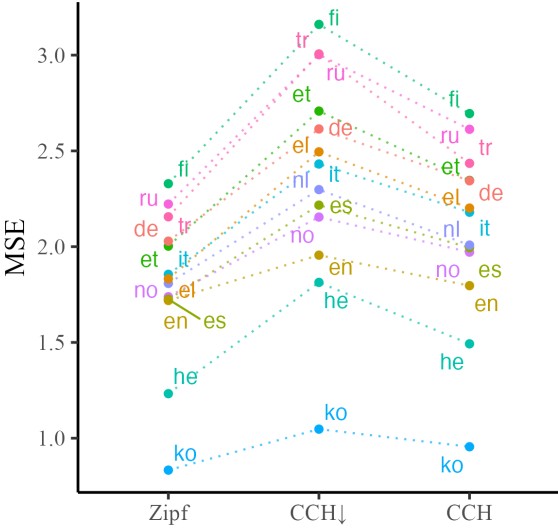

Figure 1: Mean squared error achieved by a linear model predicting real word lengths under the three hypotheses (lower is better).

In subsequent work, Piantadosi et al. (2011) offered a complementary account of communicative cost. Starting from the hypothesis that information rate should be roughly constant during communication (UID; Fenk and Fenk, 1980; Levy and Jaeger, 2007), they argue that word lengths should make information rates as close as possible to a hypothetical channel capacity, where the channel refers to the means by which information is transferred from one person to another. We term this the **channel capacity hypothesis** (CCH).[2] They conclude that lengths should be proportional to a word's **expected surprisal** instead.[3]

As the communicative efficiency of language provides important insights into human cognition (Gibson et al., 2019), Piantadosi et al.'s finding that word lengths are better explained by average surprisal than frequency has been influential. However, there are shortcomings: First, the manner in which Piantadosi et al. finds a solution which minimizes the cost associated with CCH is not formally

---

[1]We will refer to this hypothesis as ZIPF.

[2]CCH is one of the many instantiations of the uniform information density (UID) hypothesis. We introduce this new terminology here to make the hypothesis' name more descriptive.

[3]Surprisal is defined as negative log-probability in context.

specified. And second, Piantadosi et al.'s empirical results have been shown to be sensitive to a number of methodological decisions, such as the choice of text-encoding (e.g., ascii vs. unicode), the inclusion of non-conventional wordforms and other orthographic conventions of a language (Meylan and Griffiths, 2021; Levshina, 2022). Thus, there remain fundamental open questions about the relationship between communicative efficiency and word length. Here, we aim to clarify both theoretical and empirical aspects of this relationship.

Theoretically, we offer a novel, formal derivation of Piantadosi et al.'s claim. We find that Piantadosi et al. (2011) optimize not for the objective under the CCH, but for a lower bound on it instead; we call this the CCH$_\downarrow$ objective. We then provide a closed-form expression for the function that determines word lengths under CCH: Word lengths should be proportional to the **expected surprisal plus its variance-to-mean ratio**. Importantly, we derive the solution above by framing the problem of assigning wordforms as the optimization of a cost function.[4] By instantiating this optimization problem with the objectives posited by each hypothesis (ZIPF, CCH, and CCH$_\downarrow$), we can compute their word length predictions within a single, unified framework.

Empirically, we offer a large-scale comparison of ZIPF's, CCH$_\downarrow$'s, and CCH's word length predictions across 13 typologically diverse languages. Notably, we use neural language models to estimate words' surprisals, which provides more accurate estimates than the $n$-gram models relied on by prior work on this topic (Piantadosi et al., 2011; Meylan and Griffiths, 2021; Levshina, 2022). We find strong evidence (see Fig. 1) that languages are optimized to minimize their utterance lengths: A word's frequency (ZIPF's prediction) offers stronger predictive power over word lengths than either the surprisal's expected value (CCH$_\downarrow$'s prediction) or expected surprisal plus variance-to-mean ratio (CCH's prediction). We conclude that Zipf's longstanding theory stands strong.

## 2  The Lexicalization Problem

Zipf (1935, 1949) posited that the lexicon is optimized for communication, taking the needs of both speakers and listeners into account. In this section, we formalize a slice of this optimization

problem. First, we assume a fixed (but potentially infinite) vocabulary $\mathcal{W}$ of **words**, each of which we denote as $w \in \mathcal{W}$, and a fixed alphabet $\Sigma$. Given a vocabulary and alphabet, we define a **lexicon** as a function that outputs a **wordform** for each word; we denote a lexicon as $\phi : \mathcal{W} \to \Sigma^*$ and a wordform as $\phi(w) \in \Sigma^*$. Note that we distinguish between a word, which is an abstract notion or concept, and its wordform, which is its ortho-phonological realization. Further, let $p(w, c)$ be a language's joint probability distribution over these words and their prior linguistic context $c \in \mathcal{W}^*$.[5] Finally, let $\mathrm{cost}[\phi](w, c)$ be a **cost function** that, given a lexicon, outputs the communicative cost of a word in context. It is often suggested that the only attribute of a wordform $\phi(w)$ that the function $\mathrm{cost}[\phi]$ is concerned with is its length $|\phi(w)|$, where $|\cdot| : \Sigma^* \to \mathbb{Z}_+$. We now define the optimization problem proposed by Zipf as follows.

**Definition 1.** *The **lexicalization problem** is the task of finding an optimal lexicon $\phi^*$, which minimizes* $\mathrm{cost}[\phi]$. *This lexicon can be described formally as the solution to*

$$\phi^* = \operatorname*{argmin}_{\phi} \operatorname*{E}_{p(w,c)} \mathrm{cost}[\phi](w, c) \tag{1}$$
$$\textit{subject to} \quad \phi \in \Phi_\ell$$

*where $\Phi_\ell$ is the set of valid $\phi$ for language $\ell$.*

There are many assumptions that one could make about $\Phi_\ell$'s characteristics. We make a few explicit in the following remark.

**Remark 1.** *We take the set $\Phi_\ell$ to include all lexicons which:* ① *only produce phonotactically valid wordforms,*[6] ② *respect morphological composition,*[7] *and* ③ *are uniquely decodable.*[8]

Another implicit constraint ④ regarding valid $\phi$— which comes from our specification of the output

---

[4]As we will make explicit, we relax some optimization constraints to be able to derive closed-form solutions. These solutions will thus lead to lower bounds on the total cost.

[5]We define this distribution formally in App. A.

[6]Phonotactics tells us how phones can be combined to create wordforms in a language. If we denote the set of all possible phonotactically valid wordforms in language $\ell$ as $L_\ell \subset \Sigma^*$, this means that the image of $\phi$ is contained in $L_\ell$.

[7]Roughly, if the concepts represented by $w$ and $w'$ overlap in a dimension that is captured by $\ell$'s morphology (e.g., plurality in English) then their wordforms $\phi(w)$ and $\phi(w')$ are likely to also partially overlap.

[8]This condition is perhaps too strict. Homophony, for instance, is when $\phi(w) = \phi(w')$ for $w \neq w'$ and will, in general, make natural languages not uniquely decodable. However, if two words never appear in the same context, natural languages may still be uniquely decodable even in the presence of homophony. We note that whether or not natural languages are optimized for being unambiguous is contentious (Chomsky, 2002; Piantadosi et al., 2012; Pimentel et al., 2020, 2021b; Trott and Bergen, 2020, 2022).

space of $\phi$—is that these mappings only produce *integer-length* wordforms.

In the subsequent sections, we consider relaxations of eq. (1) to arrive at simple solutions regarding the lengths provided by optimal lexica. Specifically, we partially relax constraint ① and fully relax constraint ② when deriving a lexicon with minimal utterance length. Further, when deriving optimal results for both CCH and CCH$_\downarrow$, we also fully relax constraints ①, ③, and ④.[9] Note that, as in all optimization problems, removing constraints always yields a *lower bound* on the expected cost we obtain under an optimal lexicon.[10]

## 3 Revisiting Zipf's Law of Abbreviation

Zipf (1935, 1949) posited a specific form that the cost function in eq. (1) should take. Concretely, he posited that lexica were optimized with the goal of minimizing speakers' utterance lengths, which can be written as $\mathrm{cost}[\phi](w, c) = |\phi(w)|$ in our notation. In an attempt to formalize his position, he proposed his eponymous **law of abbreviation**:

$$|\phi_{\mathrm{zipf}}(w)| \propto -\log p(w) \qquad (2)$$

Over the years, Zipf's law of abbreviation has been empirically investigated numerous times (Wimmer et al., 1994; Sigurd et al., 2004; Kanwal et al., 2017; Koplenig et al., 2022; Levshina, 2022; Petrini et al., 2022, 2023). We now present a formal derivation of Zipf's law of abbreviation by viewing it as an instantiation of the lexicalization problem.

**Hypothesis 1.** *Zipf's hypothesis predicts that communication is made optimal by the mapping $\phi_{\mathrm{zipf}}$ that satisfies:*

$$\phi_{\mathrm{zipf}} = \operatorname*{argmin}_{\phi} \; \operatorname*{E}_{p(w,c)} |\phi(w)| \qquad (3)$$
$$\text{subject to} \quad \phi \in \Phi_\ell$$

If we relax constraints ① and ② in Remark 1, then the optimal solution to eq. (3) can be achieved by Huffman coding (Huffman, 1952).[11] We know that this optimal solution's word lengths respect:

$$|\phi_{\mathrm{zipf}}(w)| \leq -\log_{|\Sigma|} p(w) + 1 \qquad (4)$$

which can be roughly approximated as $|\phi_{\mathrm{zipf}}(w)| \approx -\log_{|\Sigma|} p(w)$. Unfortunately, empirical evidence suggests that this solution, which suggests the proportionality constant in eq. (2) equals 1, is *not* representative of how natural languages behave (Pimentel et al., 2021c). It thus gives us little insight into how actual wordforms should behave.

Fortunately, we can derive a more interesting result where the proportionality in eq. (2) still holds by only *partially* relaxing ① from Remark 1. We first assume a very simplistic model of phonotactics. Given an alphabet $\Sigma$ of phones, let $L_\ell \subset \Sigma^*$ be the set of phonotactically valid wordforms in language $\ell$. Note that this assumes *deterministic* phonotactics (Gorman, 2013; Dai and Futrell, 2021).[12] Further, define $\mathrm{PREFIXES}(L_\ell) \stackrel{\mathrm{def}}{=} \{\boldsymbol{\alpha}_{<t} \mid \boldsymbol{\alpha} \in L_\ell, t \leq |\boldsymbol{\alpha}|\}$ to be the set of all prefixes in this language.

**Assumption 1.** *The **constant phonotactic assumption** assumes there exists a $K \in \mathbb{Z}_{>0}$ such that $K \leq |\Sigma|$ and, for every string $\boldsymbol{\alpha} \in \mathrm{PREFIXES}(L_\ell)$, there exist* exactly $K$ *symbols $\{\sigma_k\}_{k=1}^K$ for which $\boldsymbol{\alpha}\sigma_k \in \mathrm{PREFIXES}(L_\ell)$.*

In words, Assumption 1 says that there are exactly $K$ valid symbols with which every phonotactically valid prefix can be extended. Given this assumption, we can now find a solution to eq. (3), which only partially relaxes the phonotactic constraint in Remark 1.

**Theorem 1.** *The minimization problem given in Hypothesis 1 with constraint ② relaxed can be solved by Huffman coding[13] with $K$ symbols. The optimal solution is given by*

$$|\phi_{\mathrm{zipf}}(w)| = |\phi_{\mathrm{huff}_K}(w)| \qquad (5a)$$
$$\leq -\frac{1}{\log_{|\Sigma|} K} \log_{|\Sigma|} p(w) + 1 \qquad (5b)$$

*Proof.* The proof is available in App. C. ∎

Theorem 1 makes precise the sense in which we claim to have derived Zipf's law of abbreviation. Under the rough approximation $|\phi_{\mathrm{zipf}}(w)| \approx -\frac{\log_{|\Sigma|} p(w)}{\log_{|\Sigma|} K}$, the proportionality in eq. (2) is realized through the unknown constant $1/\log_{|\Sigma|} K$.

---

[9]Explicitly, by relaxing ④, we allow $|\phi(\cdot)|$ to take on continuous values. Such a relaxation destroys $\phi$'s interpretation as assigning wordforms that live in $\Sigma^*$. However, it turns a combinatorial optimization problem into a continuous one and allows us to apply tools from calculus.

[10]Pimentel et al. (2021c) estimate eq. (1) while respecting all four constraints, but restricted to $\mathrm{cost}[\phi](w, c) = |\phi(w)|$.

[11]By Kraft-McMillan's inequality, constraining our solution to not only be uniquely decodable, but to be prefix free, adds nothing in terms of length (Cover and Thomas, 2005).

[12]Non-deterministic models of phonotactics are also popular; see (Hayes and Wilson, 2008) for a classic study.

[13]Huffman coding is an efficient $\mathcal{O}(|\mathcal{W}|\log|\mathcal{W}|)$-algorithm that returns the exact solution to this problem. Huffman coding, however, requires a finite $\mathcal{W}$, which may not always be the case in theory. Linder et al. (1997) proved the existence of an optimal code which respects eq. (4) for distributions with infinite support but finite entropy. See Pimentel et al. (2021c) for more discussion.

## 4   Revisiting Piantadosi et al. (2011)

What's wrong with Zipf's law of abbreviation? The solution in eq. (5) is only optimal if one believes that $\mathrm{cost}[\phi](w,c) = |\phi(w)|$ is the true objective underlying the lexicalization problem. However, more recent work on communicative efficiency (e.g., Piantadosi et al., 2009, 2011) has proposed that speakers may intend to optimize another objective instead. Specifically, one can take the perspective that language is an exchange of information via a noisy communication channel, where **information** is operationalized as a word's surprisal $\mathrm{H}(w \mid c) = -\log p(w \mid c)$. This channel has an inherent **capacity** $\mathfrak{C} \in \mathbb{R}_{>0}$ at which information can be transmitted while the receiver is still able to effectively decode the underlying message. Under this perspective, optimal communication happens when a word's **information rate** ($\frac{\mathrm{H}(w|c)}{|\phi(w)|}$, in bits per character) is kept as close as possible to $\mathfrak{C}$. A word's **channel deviation** is then the difference between its information rate and channel capacity. This hypothesis can thus be stated within the framework of the lexicalization problem by defining the $\mathrm{cost}[\phi](w,c)$ of a lexicon as a function of the channel deviation.

**Hypothesis 2.** *The **channel capacity hypothesis** predicts that communication is made optimal by the mapping $\phi_{\mathrm{cch}}$ that satisfies:*

$$\phi_{\mathrm{cch}} = \underset{\phi}{\mathrm{argmin}} \; \underset{p(w,c)}{\mathrm{E}} \; \mathrm{dist}\left(\frac{\mathrm{H}(w \mid c)}{|\phi(w)|}, \mathfrak{C}\right) \tag{6}$$
$$\textit{subject to} \quad \phi \in \Phi_\ell$$

*where* $\mathrm{dist}(x,y)$ *is a function that quantifies how far $x$ is from $y$.*[14]

Intuitively, eq. (6) penalizes lexica where the length of a word causes its information rate to deviate from the channel capacity. Thus, $\phi_{\mathrm{cch}}$ will generate word lengths which produce information rates that are as uniform as possible. It follows that it can be categorized under the larger umbrella of the uniform information density hypothesis (UID; Fenk and Fenk, 1980; Levy and Jaeger, 2007). As discussed by Meister et al. (2021), however, UID has several potential interpretations, only one of which involves maximizing the use of a communication channel. Here, we will only discuss it under

this perspective, and assume that its operationalization is given by eq. (6).

### 4.1   Optimal Word Lengths

The exact solution to eq. (6) depends on the choice of dist. In this section, we assume a quadratic distance function, i.e., $\mathrm{dist}(x,\mathfrak{C}) = (x - \mathfrak{C})^2$. Efficient lexica should thus minimize the expected value of the square of the channel deviation under $p(w,c)$ (i.e., its mean squared error). We now derive a closed-form expression for CCH-optimal word lengths under this cost function. As in Theorem 1, we relax the morphological ② constraint. Beyond this, we also relax the phonotactic ①, unique-decodability ③, and the integer-length ④ constraints. Note that, unlike in Theorem 1, we need to relax ④ here because we have no efficient combinatorial algorithm to solve eq. (6).

**Theorem 2.** *Under Hypothesis 2, if we relax ①, ②, ③ and ④, the optimal word lengths are given by*

$$|\phi_{\mathrm{cch}}(w)| = \frac{1}{\mathfrak{C}} \frac{\underset{p(c|w)}{\mathrm{E}}\left[\mathrm{H}^2(w \mid c)\right]}{\underset{p(c|w)}{\mathrm{E}}\left[\mathrm{H}(w \mid c)\right]} \tag{7}$$

*Proof.* The proof is available in App. D.   ∎

We note that Eq. (7) is equivalent to the expected surprisal plus a variance-to-mean ratio.[15]

### 4.2   Choices of Distance

In the above section, we assumed a quadratic penalty for a word's channel deviation. There is, however, no inherent reason why dist should be quadratic. We thus examine alternative ways to quantify the deviation between a word's information rate and the channel capacity. Different choices of dist will then each define a cost function through $\mathrm{cost}[\phi](w,c) = \mathrm{dist}\left(\frac{\mathrm{H}(w|c)}{|\phi(w)|}, \mathfrak{C}\right)$.

Any specific utterance should fall in one of three cases: First, a word's information rate may be **at capacity**, i.e., when $\frac{\mathrm{H}(w|c)}{|\phi(w)|} = \mathfrak{C}$. In this case, there are no CCH-based costs. As the capacity is a real number, however, this is virtually impossible in practice. Second, information rate may be **below capacity**. This will imply an opportunity cost on communication: speakers will need more time to produce their utterances than desired, which is not ideal from the perspective of communicative efficiency (Levy and Jaeger, 2007; Kanwal, 2018).

---

[14]We consider $\mathrm{dist}(\cdot)$ functions which satisfy the first two axioms required by true distance metrics: $\mathrm{dist}(x,y) = 0 \iff x = y$ (identity of indiscernibles) and $\mathrm{dist}(x,y) \geq 0$ (non-negativity), but which are not necessarily symmetric and do not necessarily satisfy the triangle inequality.

[15]This can be seen via the following manipulations: $\frac{\mathrm{E}[x^2]}{\mathrm{E}[x]} = \mathrm{E}[x] + \frac{\mathrm{E}[x^2] - \mathrm{E}[x]^2}{\mathrm{E}[x]} = \mathrm{E}[x] + \frac{\mathbb{V}[x]}{\mathrm{E}[x]}$

Third, information rate may be **above capacity**. This again implies a cost on communication; since communicating above a channel's capacity is provably noisy (Shannon, 1948), there may be communication faults which will either lead to the wrong meaning being conveyed, or will require a potential retransmission of the message.

The quadratic distance function that we have proposed above assumes a symmetric cost, where communication above or below capacity are equally harmful. It is, however, reasonable to assume that the cost associated with communicating above capacity may be higher than the opportunity cost of communicating below it. This leads us to propose costs based on the following generalized distance function:

$$\text{dist}(x, \mathfrak{C}) = \begin{cases} \lambda\,(x - \mathfrak{C})^2 & \textbf{if } x > \mathfrak{C} \\ (x - \mathfrak{C})^2 & \textbf{else} \end{cases} \quad (8)$$

where $\lambda \in \mathbb{R}_{>0}$. Under this generalized distance function, any value $\lambda > 1$ will imply a larger penalty for communicating above than below capacity. Further, when $\lambda = 1$ we recover the symmetric quadratic distance function proposed earlier.

Notably, when assuming this generalized distance function, there is no capacity-agnostic closed-form value to which word lengths should be proportional. Here, we find $\text{CCH}_\lambda$-optimal lengths with a two step process: (i) given a large set of surprisal values paired with their word lengths, we find what the optimal capacity is for a language; (ii) we then use a gradient descent-based optimizer to find the optimal lengths under that capacity.

### 4.3 Piantadosi et al.'s (2011) Lower Bound

In their paper, Piantadosi et al. offer a similar argument to the one proposed in this section. They state, however, that the optimal word lengths follow:

$$|\phi_{\text{cch}_\downarrow}(w, c)| \propto \text{H}(w \mid C) \quad (9)$$

where $\text{H}(w \mid C)$ is the surprisal of word $w$, marginalized over all contexts. While Piantadosi et al. intended to find a solution which minimizes the cost associated with CCH, they actually do something else. We find that Piantadosi et al.'s proposal optimizes a *different* instantiation of the lexicalization problem, one that does not use the objective that formally corresponds to the CCH hypothesis.[16] We give the objective Piantadosi et al.'s proposal is the solution to below as its own hypothesis.

---

[16]See Cohen Priva (2015), however, for a discussion on how average surprisal could still predict a word's duration beyond individual surprisal effects.

**Hypothesis 3.** *Piantadosi et al. predict that communication is made optimal by the mapping $\phi_{\text{cch}_\downarrow}$ that satisfies:*

$$\phi_{\text{cch}_\downarrow} = \operatorname*{argmin}_{\phi} \operatorname*{E}_{p(w,c)} \text{dist}\left( \frac{\text{H}(w \mid C)}{|\phi(w)|}, \mathfrak{C} \right) \quad (10)$$
$$\textit{subject to} \quad \phi \in \Phi_\ell$$

We now give the connection between Hypothesis 3 and eq. (9) in the following theorem.

**Theorem 3.** *Under Hypothesis 3, if we further relax ①, ②, ③ and ④, the optimal word lengths are given by*

$$|\phi_{\text{cch}_\downarrow}(w)| = \frac{1}{\mathfrak{C}} \text{H}(w \mid C) \quad (11)$$

*Proof.* Using $\phi = \phi_{\text{cch}_\downarrow}$ as given by eq. (11), we get $\text{dist}(\cdot, \mathfrak{C}) = 0$ for all words when evaluating the objective in eq. (10). By definition, this is the minimum for any $\text{dist}$. ∎

Note that $\text{dist}\left( \frac{\text{H}(w|C)}{|\phi(w)|}, \mathfrak{C} \right)$ is constant with respect to individual contexts $c$. Thus, the expectation in eq. (10) can simply be taken over the unigram distribution, $p(w)$. Moreover, if $\text{dist}$ is a convex function, then, we can use Jensen's inequality to show that eq. (10) lower-bounds eq. (6).[17] We therefore denote Piantadosi et al.'s hypothesis and solution $\text{CCH}_\downarrow$.

**Proposition 1.** *Given a convex $\text{dist}$ function and any $\phi \in \Phi_\ell$, the cost optimized by $\text{CCH}_\downarrow$ in Hypothesis 3 lower-bounds CCH's cost in Hypothesis 2*

$$\operatorname*{E}_{p(w,c)} \text{dist}\left( \frac{\text{H}(w \mid c)}{|\phi(w)|}, \mathfrak{C} \right)$$
$$\geq \operatorname*{E}_{p(w,c)} \text{dist}\left( \frac{\text{H}(w \mid C)}{|\phi(w)|}, \mathfrak{C} \right) \quad (12)$$

*Proof.* The proof is available in App. E. ∎

We now provide an example to show how $\text{CCH}_\downarrow$'s solution does not minimize $\text{dist}\left( \frac{\text{H}(w|c)}{|\phi(w)|}, \mathfrak{C} \right)$ under the distribution $p(w, c)$.

**Example 1.** *Let there be a word with a surprisal of 2 bits in ten distinct contexts, and a surprisal of 24 bits in a single context; assume all eleven contexts are equiprobable. The word's average surprisal is thus 4 bits (i.e., $\frac{10\cdot2+24}{11}$). Further, assume we*

---

[17]Note that this lower bound is with respect to the function being minimized in our optimization problem. It is therefore in addition to the lower bound that comes from relaxing this optimization problem's constraints.

have a channel with capacity $\mathfrak{C} = 2$. *According to Theorem 3, we have* $|\phi_{\text{cch}_\downarrow}(w)| = \frac{\text{H}(w|C)}{\mathfrak{C}} = 2$, *which under the* CCH *objective (eq. (6)) gives us an expected cost of* $10$ *(i.e.,* $\frac{10}{11}\left(\frac{2}{2}-2\right)^2 + \frac{1}{11}\left(\frac{24}{2}-2\right)^2$*). If we choose word lengths according to Theorem 2 instead, we get that the length should be* $|\phi_{\text{cch}}(w)| = 7$. *This results in a cost under the* CCH *objective of roughly* $2.86$.

## 5 Experimental Setup

### 5.1 Estimating Word Length Predictions

To evaluate the different hypotheses, we test how well their predictions about word lengths align with the lengths of real languages' wordforms. These predictions require computing surprisals (either unigram or contextual), which are defined according to the true probability distribution $p$ (either as a function of $p(w)$, or $p(w \mid c)$; the distribution $p$ is defined more precisely in App. A). While we do not have access to the true probability distribution $p$, we do have samples from it. We use the following estimators of eqs. (5), (7) and (11):

$$|\widehat{\phi_{\text{zipf}}(w)}| = -\log q(w) \tag{13a}$$

$$|\widehat{\phi_{\text{cch}_\downarrow}(w)}| = -\frac{1}{|\mathcal{D}_w|}\sum_{c' \in \mathcal{D}_w} \log q(w \mid c') \tag{13b}$$

$$|\widehat{\phi_{\text{cch}}(w)}| = -\frac{\sum\limits_{c' \in \mathcal{D}_w}\left(\log q(w \mid c')\right)^2}{\sum\limits_{c' \in \mathcal{D}_w} \log q(w \mid c')} \tag{13c}$$

where $\mathcal{D}_w = \{c' \mid (c', w') \in \mathcal{D}, w' = w\}$, and $\mathcal{D}$ is our corpus, which we assume to be sampled from the distribution $p$. In practice, our corpus $\mathcal{D}$ is composed of data from one out of 13 languages from 5 language families in Wiki40B (Guo et al., 2020).

Distribution $q$ is our estimate of $p$, which we implement using language models. We use: normalized count statistics to estimate the unigram distribution $p(w)$, and transformer models for $p(w \mid c)$. Our data and models are described in detail in App. B.[18] Note that we omit unknown constants from eqs. (13a) to (13c) because we only consider scale-invariant evaluation metrics.

---

[18]In our main set of experiments, we filter the set of words we analyze to only include the top 25k most frequent words in a language which have wordforms composed of characters in the language's alphabet; we use alphabet's as defined in homoglyph: https://pypi.org/project/homoglyphs/. We also pre-tokenize data with language-specific UnigramLM tokenizers, and sum subword surprisals when necessary to get per-word values.

### 5.2 Evaluation Metrics

Even with access to the true $p$, comparing the word length predictions of the different theories above would be non-trivial. Language evolution is a dynamic and noisy process: Even if one of the above optimization pressures has acted during the creation of languages' lexica, it is unlikely that they are perfectly optimal with respect to that pressure. We thus cannot simply evaluate whether languages match our predictions exactly. Rather, we can instead measure if the general trends predicted by the different hypotheses match the trends observed in natural language. We will rely on a number of metrics to evaluate our results. Taken together these metrics should allow us to draw conclusions on which theory (if any) best correlates with observed word lengths.

**Spearman Correlation.** First, we follow prior work (Piantadosi et al., 2011; Meylan and Griffiths, 2021; Levshina, 2022) and use the Spearman correlation to assess the quality of each word-length hypothesis. A positive attribute of this correlation is that it can account for nonlinear relationships, potentially accounting for non-linear optimization obstacles. This metric, however, has a significant drawback: Namely, all wordforms contribute equally to its computation. If we evaluate large enough corpora using Spearman correlations, we will therefore consider vocabularies $\mathcal{W}$ mostly dominated by low-frequency and uncommon wordforms, such as typos, specialized terms, and names. Yet arguably, when evaluating the different hypotheses, a given word should be weighted according to its usage (i.e, frequency) in a given language, as this is the case in our various optimization problems; a word's impact on the lexicalization problem's objective is a function of its frequency. This is perhaps one of the reasons why prior work has limited their analyses to only consider a subset of the most common words per language (Piantadosi et al., 2011), a design choice that we likewise employ in our main experiments.

**Pearson Correlation.** As a second metric, we evaluate the Pearson correlation between our predictions and actual word lengths. Pearson's correlation has similar drawbacks to Spearman's, differing from it only in that its value reflects the strength of *linear* relationships.

**Weighted Mean Squared Error (MSE).** As a third metric, we use weighted MSE, which

avoids the main drawbacks of the previous metrics. We fit linear regression models (without a bias term) to predict a language's word lengths using our ZIPF, CCH, or CCH↓ estimators as the sole predictor. Importantly, we weight each squared error term by that words' frequency (both during this model's training and evaluation). This design choice makes our method more robust to the set of words being evaluated, since the inclusion of exponentially many low-frequency words should not substantially affect weighted MSE. Note that this procedure is equivalent to measuring the predictive power of each hypothesis, while assuming eqs. (5), (7) and (11) predict an expected length, and that word lengths are normally distributed around these expected values.

## 6 Results

Our main results are presented in Fig. 1 and 2. In short, Fig. 1 shows that words' frequencies offer stronger predictive power of word lengths (as evinced by smaller MSE) than either of the surprisal-dependent metrics. This result provides evidence for ZIPF's hypothesis over either CCH or CCH↓. This result is particularly surprising since we improve on CCH's optimal word length predictions, but ZIPF's hypothesis still provides the best predictions.[19] A similar result can be seen in Fig. 2, where frequency offers the strongest correlation with lengths (in terms of both Pearson and Spearman), in all languages but English. Notably, in our results, some languages even have a *negative* correlation between the two surprisal-based measures and actual word lengths. We now turn to analyzing different methodological choices that could impact our results.

### 6.1 Sensitivity to Tokenization

The first design choice that we analyze here is the choice of tokenizer that we use to preprocess our data. As cross-entropies are necessarily larger or equal to entropies,[20] it is reasonable to expect that our language model surprisal estimates may be, on average, larger than true surprisals. While we do not know the exact per-token nature of this difference, it is conceivable that using UnigramLM tokenization could compound it: On average,

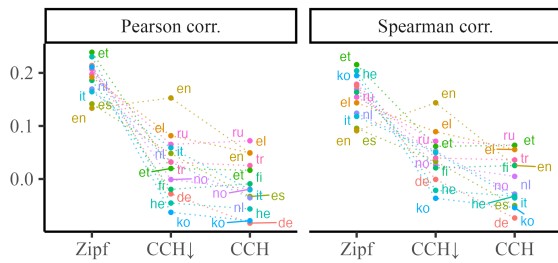

Figure 2: Pearson and Spearman correlation of the three hypotheses in the analyzed languages (higher is better).

longer words will naturally decompose into more subword units, and so when adding subword surprisals, the total error of a word's surprisal estimate may correlate with its number of subword units.

To assess the impact of this potential systematic error in our estimates, we thus re-train our models using a vocabulary of 32k *full* words, replacing any word not in this set with an unk symbol, which is necessary when working with finite vocabularies. Under this model, all analyzed words are encoded using a single "subword" unit. We then re-analyze the three hypotheses as before. In Fig. 3 (top), we see that a word's frequency is still a better predictor of its length than the quantities put forth by other hypotheses. Further, in the only case in which CCH↓ offers better predictions than ZIPF (English, as evinced by higher Spearman correlations), their performance difference is now lower than before.[21]

We also estimate ZIPF's unigram surprisals using tokenized counts, i.e., where we count subword tokens to estimate frequencies instead of directly counting the full words in our training set. We then estimate the suprisal of a word as the sum of the surprisals of its subwords, thus assuming independence between them. We display these new results in Fig. 3 (bottom) under the name **Zipf (subwords)**. We see here that this tokenization scheme increases our measured correlations, and Zipf (subwords) presents the strongest correlations in all languages. Perhaps surprisingly, tokenization seems to not influence MSE as much.

### 6.2 Sensitivity to Word Filtering Protocol

Next, we analyze our results' sensitivity with respect to how we select the set of words we analyze. Specifically, for our analyses so far we have only considered words whose wordform is composed

---

[19]We improve CCH's optimal word length predictions over prior work both theoretically, by optimizing CCH as opposed to CCH↓, and empirically, by using stronger language models.

[20]Cross-entropy is the (probability-weighted) average of the surprisal estimates from our language model.

[21]These correlations are, respectively, 0.09 vs. 0.21 with ZIPF and CCH↓ when using UnigramLM. After switching to full words they are 0.09 vs. 0.14.

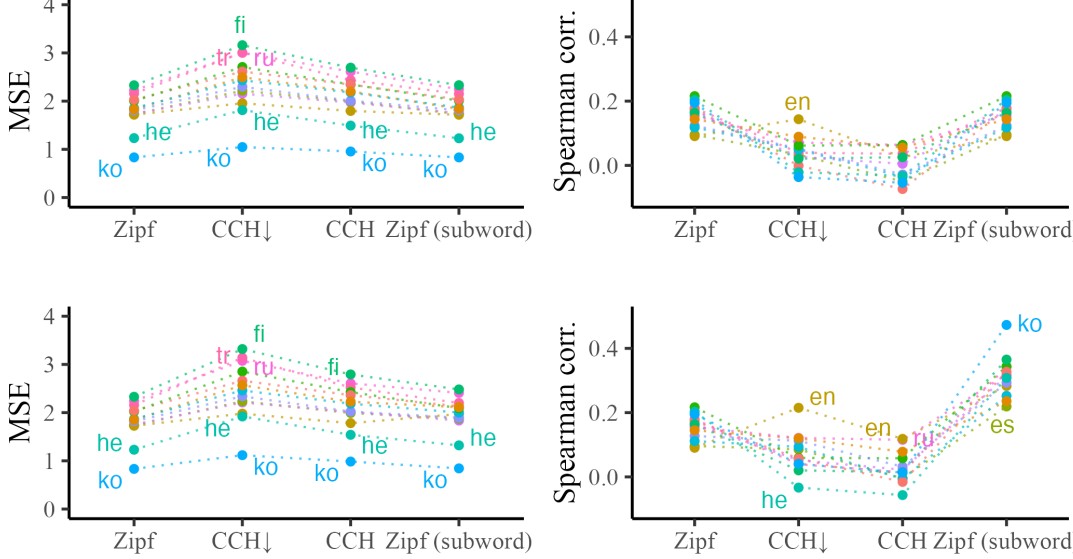

Figure 3: MSE and Spearman correlation when surprisals are estimated using either: full words directly (top), or adding subword surprisals (bottom). Note that when using full words, Zipf and Zipf (subword) are the same.

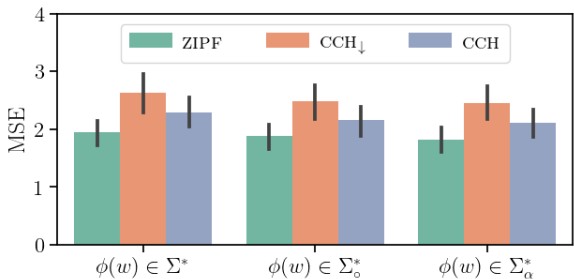

Figure 4: Average MSE across languages when hypotheses are evaluated using different word filtering protocols.

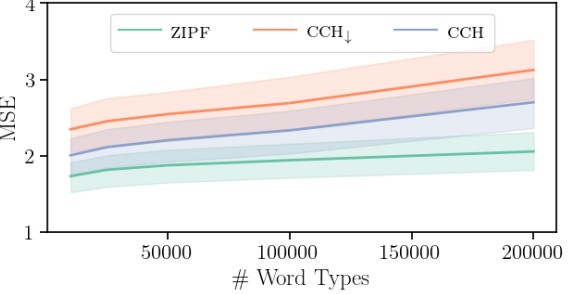

Figure 5: Average MSE across languages when hypotheses are evaluated on different number of word types.

exclusively of characters in its language's alphabet. We now run similar analyses, but including either: All white-space-separated words in a language's test set, or all white-space-separated words with no punctuation symbols.[22] We denote these conditions as: $\Sigma_\alpha^*$ when selecting alphabet-only words, $\Sigma_o^*$ when selecting no-punctuation words, and $\Sigma^*$ when selecting all words. We display results under each condition in Fig. 4. We see that across these various protocols, ZIPF's hypothesis remains the most predictive.[23]

Additionally, we consider the impact of including only the top 25k most frequent words in our analysis. In Fig. 5, we present MSE values computed when using sets composed from the top 10k most frequent words, to entire test sets. Notably, we again see that frequency remains the best pre-

---

[22]We consider punctuation to be any of: !"#$%&'()*+,-./:;<=>?@[\]^_'{|}~.

[23]In App. G's Fig. 9, we also see that Spearman correlation is considerably more sensitive to filtering protocols than MSE.

dictor of word length. In App. G's Fig. 10, we display results per language for MSE and Spearman correlation. There, we see that MSE rates frequency best on all languages and across all evaluated setups. Spearman correlation evaluated on few word types similarly rates frequency over CCH or CCH↓ predictions (again, except on English). When evaluated on full test-sets, Spearman correlation shows a less straightforward conclusion: While ZIPF still achieves the highest correlation in most languages, CCH↓ achieves stronger correlations in Italian, Spanish and Russian. At this stage, however, the evaluated sets are dominated by low-frequency words, which may not be representative of the evaluated languages.

## 6.3 Sensitivity to Model Quality

Finally, we investigate how our model quality influences our results. We train new models on subsets of our training sets to get language models

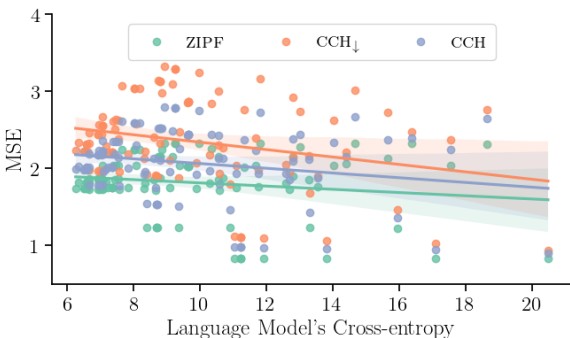

Figure 6: MSE correlation as a function of the cross-entropy of models used to get surprisal estimates.

of different qualities. We then use these models to assess whether there is a relationship between model quality and a hypothesis' predictive power. In addition to the models estimated using the full training sets, we thus train 7 new transformer and unigram models per language, each using from 1 million to 1 billion training tokens in log-uniform intervals. We plot the predictive power of each hypothesis (ZIPF's, CCH↓'s and CCH) vs. the language model's cross-entropy in Fig. 6.[24] Unintuitively, surprisal estimates of better models (i.e., with lower cross-entropies) provide *worse* predictors of word length. An additional analysis suggests that the surprisal estimates of worse language models are more strongly correlated with frequency (see Fig. 7 in App. G), which may justify this unituitive result since frequencies are most predictive of word lengths in our experiments. ZIPF's hypothesis, on the other hand, is robust to the quality of the used unigram model.

### 6.4 Sensitivity to Cost Function

In our last set of experiments, we analyze the impact of our choice of quadratic cost function in our results. Using the generalized cost function in eq. (8), we derive optimal word length predictions using values of $\lambda$ from 1 to 5 in 0.25 intervals. We present their MSE and Spearman correlations in App. G's Fig. 13. While there seems to be a slight tendency for CCH to be more predictive for larger values of $\lambda$, ZIPF still has the most predictive power of the different hypotheses.

---

[24]We show per-language plots evaluated with both MSE and Spearman correlation in App. G's Fig. 11. We do not quantify our unigram models' quality, but assume that they increase monotonically with the size of the corpus on which they were estimated. We show a similar plot, but with the number of training tokens on the $x$-axis, in App. G's Fig. 12.

## 7 Discussion

The answer to what drives the distribution of word lengths in lexica has long been considered important for understanding the evolution and function of language (see Gibson et al., 2019 for a review). Across multiple languages and various methodological choices, our results support Zipf's law of abbreviation over other potential explanations as a driving factor in the development of lexica.

These findings deviate from Piantadosi et al., who found average surprisal to be a stronger predictor of word lengths. We hypothesize that this is because of methodological choices. Specifically, Piantadosi et al. derive surprisal estimates from language models that are now outdated (in terms of their quality), and we found that, when CCH's predictions were computed using *worse* surprisal estimates, they had stronger correlations with length than when using better estimates. Like prior work on this topic (Meylan and Griffiths, 2021; Levshina, 2022), our analyses suggest the sensitivity of Piantadosi et al.'s results to methodological choices.

What do these results tell us about the communicative optimization of natural language lexica? In short, our results suggest lexica are optimized to minimize expected utterance lengths. Notably, other linguistic properties may be optimized towards other notions of communicative efficiency. While a word's duration is mainly determined by its wordform, speakers can still modulate this duration to a certain extent; such a modulation could target CCH. In fact, prior work has shown a correlation between surprisal and duration (Bell et al., 2003; Aylett and Turk, 2004; Pimentel et al., 2021a).

## 8 Conclusion

In this paper, we formalize the problem of assigning wordforms based on different notions of communicative efficiency, which we term the **lexicalization problem**. Under this framework, we describe the optimization problem related to the channel capacity hypothesis, and, in doing so, we show that Piantadosi et al.'s predictions optimized for only a lower bound on CCH, rather than on the true objective. Further, while considering relaxed versions of the lexicalization problem, we derive optimal word length values for Zipf's hypothesis and CCH. We then empirically evaluate CCH's, CCH↓'s and ZIPF's predictions in 13 languages. Our results strongly support ZIPF's hypothesis: Word lengths are optimized to minimize utterance lengths.

## Limitations

A limitation of our work is that, when deriving optimal word lengths under CCH and CCH↓, we relax: the phonotactic ①, morphological composition ②, unique decodability ③ and the integer-length ④ requirements. In the case of ③, if a language's channel capacity is large, this might lead to poorer predictions under both these theories. Deriving optimal word lengths while considering this constraint is left as an open problem for future work. In the case of ④, it is arguably unrealistic to consider continuous-length wordforms. This issue could be addressed by using a linear program to solve problems of the form eq. (1). This, as well as considering the role of phonotactics ① and morphological composition ② in CCH, is likewise left for future work. Further, we note that while we relax all four constraints to derive CCH- and CCH↓-optimal word lengths, we only relax ② (and partially ①) to derive ZIPF-optimal lengths. This could realistically impact the fact that Zipf's hypothesis seems to have more predictive power over word lengths.

Another limitation is that our analyses focus solely on written data from Wikipedia. We recommend future work investigates how these findings generalize to spoken or signed languages, and to other text genres. Finally, while we use a typologically diverse sample of languages, it is still skewed towards Eurasian languages. This is because the large amount of text needed for training state-of-the-art language models—necessary to estimate entropy—are not available in many languages. Expanding the set of languages analyzed here would be necessary to confirm the generality of our results.

## Acknowledgements

We thank the anonymous reviewers and meta-reviewer for their feedback on this paper. Tiago Pimentel also thanks Hope McGovern and Simone Teufel for helpful comments on different stages of writing this manuscript. Tiago Pimentel is funded by a Facebook PhD Fellowship. Clara Meister is funded by a Google PhD Fellowship. Ethan Gotlieb Wilcox would like to acknowledge support from an ETH Postdoctoral Fellowship.

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

## A  Defining $p(w, c)$

In this section, we explicitly define $p(w, c)$. We do this in terms of a more standard notation in language modeling. We define a sequence of words as $\boldsymbol{s} \in \mathcal{S}$ where $\mathcal{S} \stackrel{\text{def}}{=} \mathcal{W}^* \circ \{\text{eos}\}$. We then assume a distribution over such sequences $p(\boldsymbol{s})$. We can now define $p(w, c)$ as:

$$p(w, c) \propto \sum_{\boldsymbol{s} \in \mathcal{S}} p(\boldsymbol{s}) \sum_{t=1}^{T} \mathbb{1}\{w = s_t, c = \boldsymbol{s}_{<t}\} \quad (14)$$

In words, a word–context pair is as frequent as it would be in natural language, where once a sequence is uttered, all its word–context pairs are observed jointly. This is not the only possible definition of $p(w, c)$, but it is the one we opt for here.

Note that the distribution $p(w, c)$ might thus not be well defined for all distributions $p(\boldsymbol{s})$, as the normalizing constant in this definition might diverge. For instance, this distribution will not be well-defined for a language model over alphabet $\mathcal{W} = \{a\}$, where $p(a^n) = \frac{6}{\pi^2 n^2}$ for $n \geq 1$ and $p(\varepsilon) = 0$, as its sequences' average length diverges.

## B  Data and Models

**Data.**  The corpora used throughout our analyses come from Wiki40B (Guo et al., 2020). This dataset is composed of cleaned text from Wikipedia articles in more than 40 languages, out of which we select a subset of 13 for our analysis. Our selection includes: German, Greek, English, Spanish, Estonian, Finnish, Hebrew, Italian, Korean, Dutch,

Norwegian, Russian, and Turkish. These span five language families: Afro-Asiatic, Indo-European, Koreanic, Turkic, and Uralic. The data for each language comes pre-split into a training, validation and test set. We fit our models using the first two sets, while performing our analyses exclusively on the test-sets. As discussed above, the set of analyzed words may make a large difference in the measured correlations. In our main set of experiments, we filter the set of words we analyze to only include wordforms composed of characters in the language's alphabet.[25] Table 1 (in App. F) includes the number of word types and tokens used per language in our analyses.

**Models.**  To estimate the unigram distribution $p(w)$, we use a simple MLE estimator: the (normalized) count statistics from our training set. To estimate contextual probabilities $p(w \mid c)$, we use an autoregressive language model $p_\theta$. Specifically, we train monolingual transformers in each language using fairseq (Ott et al., 2019) with its default language modeling hyper-parameters. Our transformers (Vaswani et al., 2017) have 6 layers, a hidden size of 512, and 8 attention heads per layer. Further, they can attend to a context size of at most 512 tokens, and we train them with a dropout of $0.1$, and a batch size of 64. We optimize our models using Adam (Kingma and Ba, 2015) with a learning rate of $5 \times 10^{-4}$, weight decay of $0.01$, and 4k warmup steps. In our main set of experiments, we further pre-tokenize each language's text using language-specific tokenizers fit (using the UnigramLM algorithm; Kudo, 2018) on their respective training sets, with a vocabulary of 32k subword units. We then compute per-word surprisals by adding the surprisals of all the subwords that the word is composed of. (We also consider other tokenization schemes, as described in §6.1.)

## C  Proof of Theorem 1

Before proving Theorem 1, we provide a lemma which will be useful for it. In words, we prove a length-preserving bijection between $L_\ell$ and $\Delta^*$ for an alphabet $K$ such that $|\Delta| = K$.

**Lemma 1.** *Under the constant phonotactic assumption, there exists an alphabet $\Delta$ with cardinality $K$ such that, for every $N \geq 0$, $\Delta^N$ is isomorphic to $L_\ell^{(N)}$, where $L_\ell^{(N)}$ is the set of*

---

[25]We use alphabet's as defined in homoglyph: https://pypi.org/project/homoglyphs/.

*phonotactically valid wordforms with length $N$.*

*Proof.* First, it is clear that $|\Delta^N| = |\Delta|^N = K^N$. We now prove the same for $L_\ell^{(N)}$ by induction.

**Base case ($N = 0$).** The set of 0-length phonotactically valid strings includes only the empty string $\{\varepsilon\}$. It follows that: $|L_\ell^{(0)}| = 1 = K^0$.

**Inductive step ($N > 0$).** By the inductive hypothesis, we have that $|L_\ell^{(N-1)}| = K^{N-1}$. By Assumption 1, each element in $L_\ell^{(N-1)}$ has $K$ possible continuations in $L_\ell^{(N)}$. It follows that $|L_\ell^{(N)}| = |L_\ell^{(N-1)}| K = K^N$.

Since $\Delta^N$ and $L_\ell^{(N)}$ have the same number of elements for every $N \geq 0$, there exists an isomorphism between them. ∎

Given the lemma above, we are now in a position to prove Theorem 1.

**Theorem 1.** *The minimization problem given in Hypothesis 1 with constraint ② relaxed can be solved by Huffman coding with $K$ symbols. The optimal solution is given by*

$$|\phi_{\mathrm{zipf}}(w)| = |\phi_{\mathrm{huff}_K}(w)| \tag{5a}$$

$$\leq -\frac{1}{\log_{|\Sigma|} K} \log_{|\Sigma|} p(w) + 1 \tag{5b}$$

*Proof.* Since $L_\ell^{(N)}$ is isomorphic to $\Delta^N$ for every $N \geq 0$, there exists a length-preserving bijection $\psi$ between $L_\ell$ and $\Delta$ (by Lemma 1). By Huffman's (1952) algorithm, we can construct an encoding that satisfies

$$|\psi(\phi_{\mathrm{zipf}}(w))| \leq -\log_{|\Delta|} p(w) + 1 \tag{15}$$

However, because $\psi$ is length-preserving, $|\psi(\phi_{\mathrm{zipf}}(w))| = |\phi_{\mathrm{zipf}}(w)|$. As an upper bound, we thus have

$$|\phi_{\mathrm{zipf}}(w)| \leq -\log_{|\Delta|} p(w) + 1 \tag{16a}$$

$$= -\frac{1}{\log_{|\Sigma|} K} \log_{|\Sigma|} p(w) + 1 \tag{16b}$$

∎

## D  Proof of Theorem 2

**Theorem 2.** *Under Hypothesis 2, if we relax ①, ②, ③ and ④, the optimal word lengths are given by*

$$|\phi_{\mathrm{cch}}(w)| = \frac{1}{\mathfrak{C}} \frac{\mathrm{E}_{p(c|w)}\left[\mathrm{H}^2(w \mid c)\right]}{\mathrm{E}_{p(c|w)}[\mathrm{H}(w \mid c)]} \tag{7}$$

*Proof.* We can easily derive these optimal word lengths from eq. (6) by taking its derivative with respect to a specific word's length, and setting it to zero. First, we rewrite it for mathematical convenience as:

$$\mathrm{E}_{p(w)} \mathrm{E}_{p(c|w)} \left(\frac{\mathrm{H}(w \mid c)}{|\phi(w)|} - \mathfrak{C}\right)^2 \tag{17}$$

where we make the quadratic cost function explicit. We note this function is convex, and so if we find a point where its derivative is zero, we also find its global minimum. We now take its derivative with respect to a specific word's length $|\phi(w)|$ and set this derivative to zero:

$$p(w) \mathrm{E}_{p(c|w)} \left[2\left(\frac{\mathrm{H}(w \mid c)}{|\phi(w)|} - \mathfrak{C}\right) \frac{\mathrm{H}(w \mid c)}{|\phi(w)|^2}\right] = 0 \tag{18}$$

where we note that all terms involving other words will have derivative zero (with respect to this specific word $w$'s length). As the expectation is a linear operation, we can rewrite this equation as:

$$\mathrm{E}_{p(c|w)} \left[\frac{\mathrm{H}^2(w \mid c)}{|\phi(w)|^3}\right] = \mathrm{E}_{p(c|w)} \left[\mathfrak{C} \frac{\mathrm{H}(w \mid c)}{|\phi(w)|^2}\right] \tag{19}$$

Note that both the length and capacity are constant with respect to the expectation over contexts. Isolating the length term, thus, we get:

$$|\phi(w)| = \frac{1}{\mathfrak{C}} \frac{\mathrm{E}_{p(c|w)}\left[\mathrm{H}^2(w \mid c)\right]}{\mathrm{E}_{p(c|w)}[\mathrm{H}(w \mid c)]} \tag{20}$$

This completes the proof. ∎

## E  Proof of Proposition 1

**Proposition 1.** *Given a convex* dist *function and any $\phi \in \Phi_\ell$, the cost optimized by* CCH↓ *in Hypothesis 3 lower-bounds* CCH*'s cost in Hypothesis 2*

$$\mathrm{E}_{p(w,c)} \mathrm{dist}\left(\frac{\mathrm{H}(w \mid c)}{|\phi(w)|}, \mathfrak{C}\right)$$
$$\geq \mathrm{E}_{p(w,c)} \mathrm{dist}\left(\frac{\mathrm{H}(w \mid C)}{|\phi(w)|}, \mathfrak{C}\right) \tag{12}$$

*Proof.* It can be easily shown by Jensen's inequal-

ity that for any choice of $\phi$:

$$\underset{p(w,c)}{\mathrm{E}} \; \mathrm{dist}\left(\frac{\mathrm{H}(w \mid c)}{|\phi(w)|}, \mathfrak{C}\right) \tag{21a}$$

$$= \underset{p(w)}{\mathrm{E}} \underset{p(c|w)}{\mathrm{E}} \; \mathrm{dist}\left(\frac{\mathrm{H}(w \mid c)}{|\phi(w)|}, \mathfrak{C}\right) \tag{21b}$$

$$\geq \underset{p(w)}{\mathrm{E}} \; \mathrm{dist}\left(\frac{\underset{p(c|w)}{\mathrm{E}}[\mathrm{H}(w \mid c)]}{|\phi(w)|}, \mathfrak{C}\right) \tag{21c}$$

$$= \underset{p(w)}{\mathrm{E}} \; \mathrm{dist}\left(\frac{\mathrm{H}(w \mid C)}{|\phi(w)|}, \mathfrak{C}\right) \tag{21d}$$

which completes the proof. ∎

## F   Data Statistics

We provide dataset statistics in Table 1.

## G   Further Results

For a more detailed reading, we provide MSE and Spearman correlation plots similar to Fig. 1 and 2's but as bar plots in Fig. 8. We also provide per-language results:

- as a function of the word filtering protocol used in our analysis in Fig. 9;

- as a function of the number of word types included in our analysis in Fig. 10;

- as a function of our language model's cross-entropy in Fig. 11; and

- as a function of the number of tokens used to train our language models and to get word count statistics in Fig. 12.

We also provide results when CCH is defined using generalized distfunctions, i.e., for several values of $\lambda$, in Fig. 13. Finally, we show the Spearman correlation between CCH and CCH$_\downarrow$ and unigram surprisal as a function of the used language model's quality in Fig. 7.

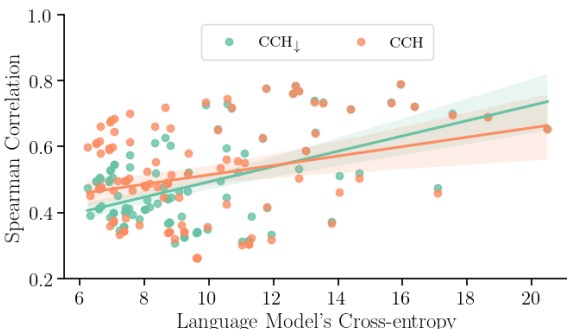

Figure 7: Spearman correlation with unigram surprisal as a function of the cross-entropy of models used to get surprisal estimates

| Language | Family | ISO code | BPC | None | | No Punctuation | | Only in Alphabet | |
|---|---|---|---|---|---|---|---|---|---|
| | | | | # Types | # Tokens | # Types | # Tokens | # Types | # Tokens |
| German | Indo-European | de | 0.99 | 2,093,524 | 32,142,917 | 1,027,594 | 27,565,045 | 896,752 | 26,301,145 |
| Greek | Indo-European | el | 1.06 | 267,625 | 2,244,964 | 145,073 | 1,954,950 | 120,361 | 1,862,380 |
| English | Indo-European | en | 1.09 | 2,419,694 | 78,392,487 | 748,109 | 66,881,077 | 609,839 | 65,261,138 |
| Spanish | Indo-European | es | 1.04 | 993,894 | 21,472,091 | 380,407 | 18,852,688 | 332,668 | 18,458,823 |
| Estonian | Uralic | et | 1.23 | 250,860 | 999,296 | 145,003 | 797,817 | 123,863 | 736,860 |
| Finnish | Uralic | fi | 1.03 | 566,755 | 2,741,783 | 333,229 | 2,249,003 | 318,514 | 2,156,132 |
| Hebrew | Afro-Asiatic | he | 1.41 | 497,550 | 4,153,846 | 230,867 | 3,394,959 | 208,077 | 3,299,727 |
| Italian | Indo-European | it | 1.04 | 823,397 | 14,500,421 | 297,153 | 12,371,701 | 269,005 | 12,056,925 |
| Korean | Koreanic | ko | 2.40 | 538,093 | 1,953,812 | 385,948 | 1,622,552 | 331,060 | 1,460,230 |
| Dutch | Indo-European | nl | 1.00 | 534,101 | 6,811,124 | 246,458 | 5,989,335 | 227,466 | 5,768,607 |
| Norwegian | Indo-European | no | 1.14 | 325,983 | 2,672,869 | 176,089 | 2,318,082 | 164,401 | 2,253,985 |
| Russian | Indo-European | ru | 1.06 | 1,474,777 | 15,824,324 | 707,806 | 13,073,143 | 546,179 | 11,922,147 |
| Turkish | Turkic | tr | 1.14 | 285,988 | 1,705,030 | 163,352 | 1,371,183 | 148,522 | 1,306,751 |

Table 1: Wiki40B data statistics.

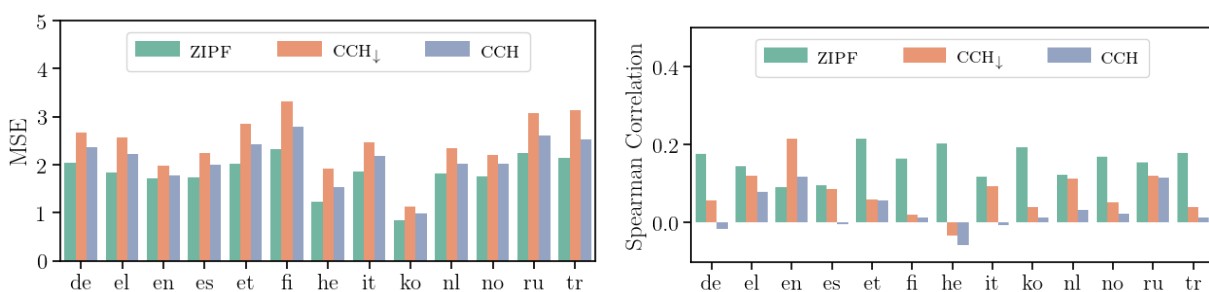

Figure 8: MSE and Spearman correlation of the three hypotheses in the analyzed languages.

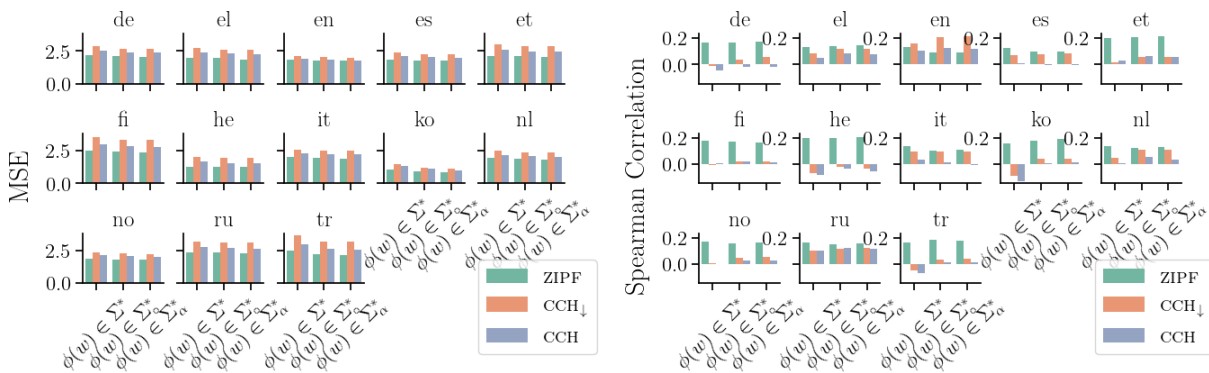

Figure 9: MSE and Spearman correlation when hypotheses are evaluated while filtering test set words based on different protocols.

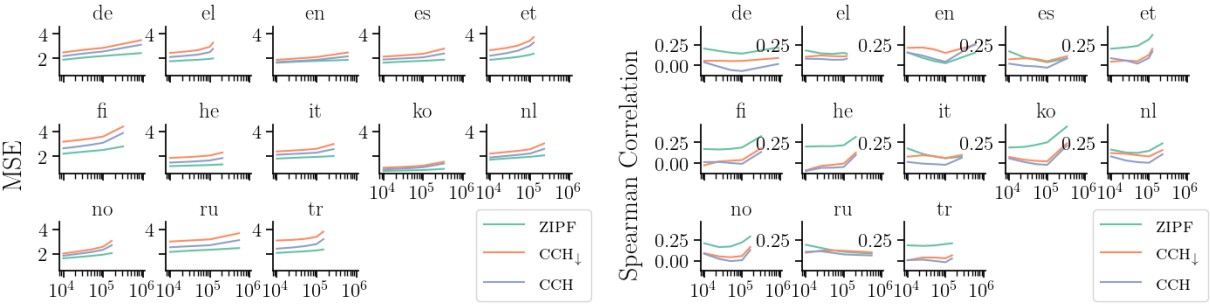

Figure 10: MSE and Spearman correlation when hypotheses are evaluated on different number of word types.

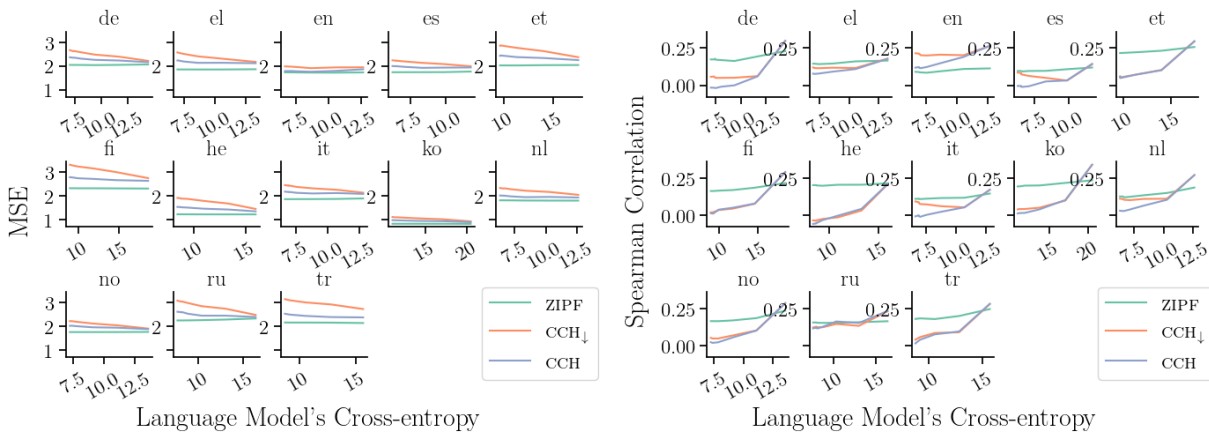

Figure 11: MSE and Spearman correlation as a function of the cross-entropy of models used to get surprisal estimates.

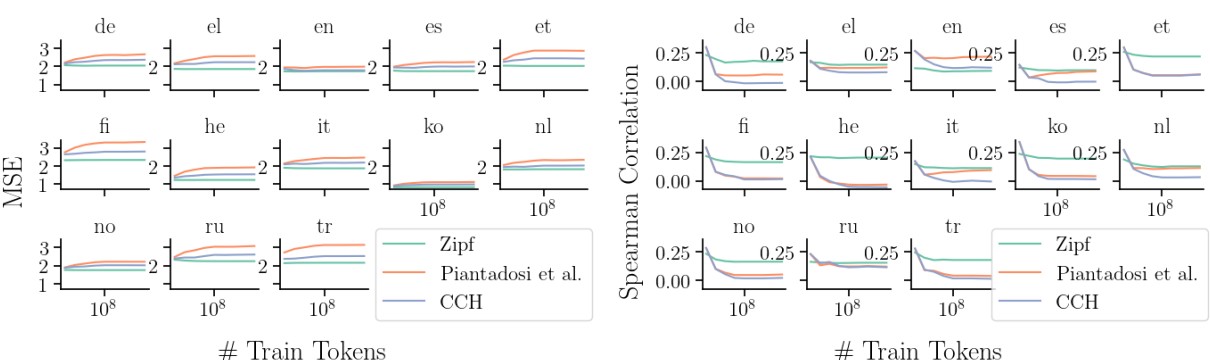

Figure 12: Results as a function of the number of tokens used to train language models and get count statistics.

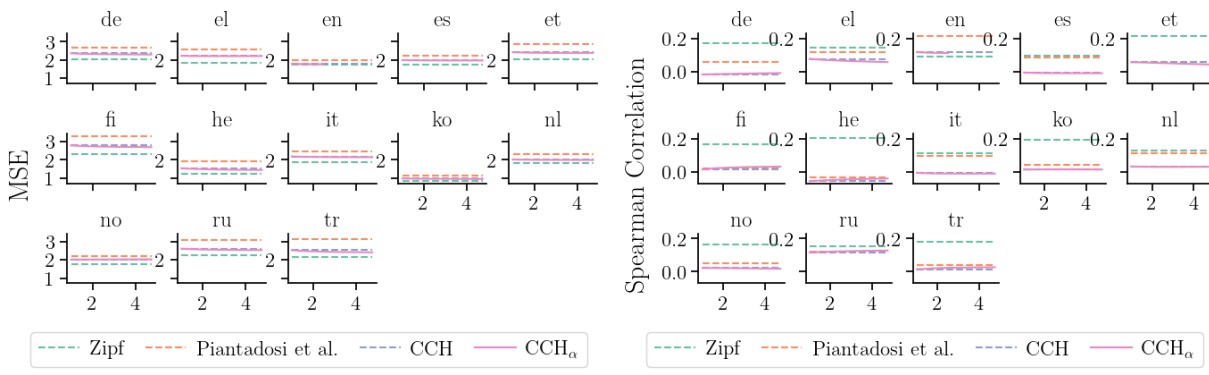

Figure 13: MSE and Spearman correlation as a function of the generalized dist's parameter $\lambda$ (in the $x$-axis).