# OpenReview forum: "Revisiting the Optimality of Word Lengths"
_EMNLP/2023/Conference — EMNLP 2023 Main_

### Official Review · Reviewer_Vf5C · 2023-08-03

**Typos Grammar Style And Presentation Improvements:** page 7 footnote
**Soundness:** 4

**Excitement:**

5: Transformative: This paper is likely to change its subfield or computational linguistics broadly. It should be considered for a best paper award. This paper changes the current understanding of some phenomenon, shows a widely held practice to be erroneous in someway, enables a promising direction of research for a (broad or narrow) topic, or creates an exciting new technique.

**Missing References:**

L 491-494 This idea comes from Menzerath-Altmann's law "The longer the whole, the smaller the parts", would be great to add a citation Menzerath 1954, Altmann 1980

**Paper Topic And Main Contributions:**

The paper addresses the well-known Zipf's law abbreviation and the ways of predicting word length in languages. In particular, the authors revisit a paper by Piantadosi et al. 2011, which claimed that word length is predicted better by surprisal than by frequency, as stated by Zipf's law. The authors find out that Zipf's law holds and that word frequency has a better predictive power than surprisal. In addition to that, the authors clarify the hypothesis by Piantadosi et al. 2011 and construct a more exact mathematical formula for their hypothesis (minimising the channel capacity cost).

**Questions For The Authors:**

A: L 204-205 "In order to solve for the optimal word lengths according to hypothesis 1, we will relax the wordform uniqueness condition" -- Can you explain why it was not taken into account in the other equations, but here this condition was relaxed? We know that in addition to phonotactics, languages tend to have quite some homography and homophony (in some languages, like French, homophony is rather high). I see that you mention this as a limitation in the end of the paper, but I wonder why sometimes this condition can be relaxed with no problem.

B: L 218-219 It would be great to see an explanation to what "index of dispersion" means

C: L 454 It's not very clear what was the task for the model, a standard masked language modelling task?

D: L 461 When using a vocabulary of subword units set to 32,000 the actual subwords will be pretty long and will be mostly similar to the actual words. This probably influences your results presented in Fig 3, thus, making both plots similar to each other. What would be interesting, is to compare several subword tokenisations with different vocabularies as a goal, e.g. 500, 1000, 15000 and to see if the metrics will change; I'd expect to see an impact of small vocabularies; when subword units would be smaller in length, they would have lower surprisal on average compared to full words constructed out of them (an extreme case would be to have just 1-character subwords, i.e. letters, to test this).

In this context, also the effects of non-alphabetic scripts would be interesting to discuss, since 1 character in English is different from 1 character in e.g. Korean. Note that Korean is located higher in Fig 3 on the bottom in Zipf (subwords) setting, but this was not discussed in the paper. Also Korean an Hebrew behave different from other languages in Fig 8 (Korean on the left, and Hebrew on the right). I would expect languages like Japanese and Chinese behave differently as well, this would be great to add to the discussion of your methods.

**Reasons To Accept:**

- The paper might have a big impact on the study of language complexity
- The paper revisits a well-known and influential hypothesis by Piantadosi et al. 2011
- The results can influence research on interpretability in multilingual NLP
- The paper is well-written, provides many experiments confirming the predictive power of word frequency


**Reasons To Reject:**

- There are small points that could be improved, which I point out below in the section Questions for the authors; otherwise many questions I had in mind were already replied by the authors in the paper

**Reproducibility:**

4: Could mostly reproduce the results, but there may be some variation because of sample variance or minor variations in their interpretation of the protocol or method.

**Reviewer Confidence:**

4: Quite sure. I tried to check the important points carefully. It's unlikely, though conceivable, that I missed something that should affect my ratings.

---

> ### Author Rebuttal · Authors · 2023-08-28
>
> We thank the reviewer for their thoughtful review! We answer the reviewer's questions below.
>
>
> > A: L 204-205 "In order to solve for the optimal word lengths according to hypothesis 1, we will relax the wordform uniqueness condition" -- Can you explain why it was not taken into account in the other equations, but here this condition was relaxed?
>
>
> We relaxed the assumption there because we were not able to derive a closed form solution to the channel capacity hypothesis (in equation 4) while considering it. We are currently still trying to find an efficient way of computing a solution to equation 4 while considering this assumption, but we were not yet able to. We will make the reason behind this limitation explicit in the paper for camera ready.
>
>
> > B: L 218-219 It would be great to see an explanation to what "index of dispersion" means
>
>
> We will add this definition to the paper: the index of dispersion is defined as the ratio between a random variable’s variance and its mean. We will also show how eq. 5 relates to it. Explicitly, we will add the equations below to make it easier for readers to follow: $\frac{\mathrm{E}[\mathrm{H}^2]}{\mathrm{E}[\mathrm{H}]} = \frac{\mathrm{E}[\mathrm{H}^2]}{\mathrm{E}[\mathrm{H}]} + \mathrm{E}[\mathrm{H}] - \mathrm{E}[\mathrm{H}] = \frac{\mathrm{E}[\mathrm{H}^2] - \mathrm{E}^2[\mathrm{H}]}{\mathrm{E}[\mathrm{H}]} + \mathrm{E}[\mathrm{H}] = \frac{\mathrm{Var}[\mathrm{H}]}{\mathrm{E}[\mathrm{H}]} + \mathrm{E}[\mathrm{H}]$.
>
>
>
> > C: L 454 It's not very clear what was the task for the model, a standard masked language modelling task?
>
>
> We train our models as autoregressive language models (line 453), so to predict a word given previous ones. We will try to make this clearer in the paper for camera ready.
>
>
> > D: L 461 When using a vocabulary of subword units set to 32,000 the actual subwords will be pretty long and will be mostly similar to the actual words. This probably influences your results presented in Fig 3, thus, making both plots similar to each other. What would be interesting, is to compare several subword tokenisations with different vocabularies as a goal, e.g. 500, 1000, 15000 and to see if the metrics will change; I'd expect to see an impact of small vocabularies; when subword units would be smaller in length, they would have lower surprisal on average compared to full words constructed out of them (an extreme case would be to have just 1-character subwords, i.e. letters, to test this).
>
>
> We agree with the reviewer that these would indeed be interesting experiments. The training of all language models in the paper was quite expensive already, though. We will consider whether to add this extra experiment for camera ready, or to leave it for future work.
>
>
> > In this context, also the effects of non-alphabetic scripts would be interesting to discuss, since 1 character in English is different from 1 character in e.g. Korean. Note that Korean is located higher in Fig 3 on the bottom in Zipf (subwords) setting, but this was not discussed in the paper. Also Korean and Hebrew behave different from other languages in Fig 8 (Korean on the left, and Hebrew on the right). I would expect languages like Japanese and Chinese behave differently as well, this would be great to add to the discussion of your methods.
>
>
> This is another good point. Non-alphabetic scripts should lead to shorter words, on average, which in turn may lead to the lower MSE values observed for, e.g. Korean, in our results. We will add a discussion on how a language’s script may influence our results for camera ready.
>
>
> > References + Typos
>
>
> We will add the suggested reference and fix the pointed out typo. Thanks for bringing them to our attention.

---

### Official Review · Reviewer_Wi23 · 2023-08-04

**Soundness:** 4

**Excitement:**

4: Strong: This paper deepens the understanding of some phenomenon or lowers the barriers to an existing research direction.

**Missing References:**

None

**Paper Topic And Main Contributions:**

This paper starts from existing theories on which information helps in previewing the length of word forms and aims at verifying them by exploiting more recent and precise language models. The considered hypotheses are introduced and carefully discussed, by also introducing new notation where necessary. In particular, they consider the Zipf law,  the channel capacity hypothesis and one variation of the latter. Empirical assessment has been performed on 13 different languages taken from the data set Wiki40B by means of an unbiased MLE estimator for unigram and monolingual transformers for the conditional probability. Results are complete and show the superiority of the Zipf law in explaining the data. In particular, the effect of different filters applied to tokenization and to the choice of the words to consider on one side and of the model quality on the other has been analyzed.

**Questions For The Authors:**

None

**Reasons To Accept:**

-- interesting problem

-- paper clear and well organized

-- convincing experimental assessment and analysis of the results

-- the paper has a few limitations, which are discussed in the last section, but the paper is interesting even with such problems

**Reasons To Reject:**

None

**Reproducibility:**

4: Could mostly reproduce the results, but there may be some variation because of sample variance or minor variations in their interpretation of the protocol or method.

**Reviewer Confidence:**

3: Pretty sure, but there's a chance I missed something. Although I have a good feel for this area in general, I did not carefully check the paper's details, e.g., the math, experimental design, or novelty.

**Typos Grammar Style And Presentation Improvements:**

Expression 4: I find that denoting context and capacity with such similar C's can be misleading: wouldn't it be possible to find a clearer notation?

line 424: hypotheses -> hypothesis

---

> ### Author Rebuttal · Authors · 2023-08-28
>
> We thank the reviewer for their thoughtful review!
>
> We will fix the typo in line 424 for camera ready, and we will try to find more distinguished notations for the context and capacity in our equations.

---

### Official Review · Reviewer_zBRX · 2023-08-04

**Soundness:** 5

**Excitement:**

5: Transformative: This paper is likely to change its subfield or computational linguistics broadly. It should be considered for a best paper award. This paper changes the current understanding of some phenomenon, shows a widely held practice to be erroneous in someway, enables a promising direction of research for a (broad or narrow) topic, or creates an exciting new technique.

**Paper Topic And Main Contributions:**

This paper explores the classical question of word length and its correlation with frequency. This classical results has been challenged by Piantadosi and colleagues, proposing a correlation between length and word's surprisal. In a first step, the paper precises the question of what is minimised in  Piantadosi's proposal, leading to a the definition of a channel capacity hypothesis (a channel corresponding information transfer between interlocutors). The paper aims at exploring the question of the  relation between communicative cost and word length and provides  theoretical and empirical contributions, providing new elements arguing in favor of the initial Zipf's hypothesis, corrrelating word lengths and frequency.

**Reasons To Accept:**

This paper is very clear and well written. It presents a very important contribution to language processing and should be definitely accepted.

**Reasons To Reject:**

No

**Reproducibility:**

5: Could easily reproduce the results.

**Reviewer Confidence:**

4: Quite sure. I tried to check the important points carefully. It's unlikely, though conceivable, that I missed something that should affect my ratings.

---

> ### Author Rebuttal · Authors · 2023-08-28
>
> We thank the reviewer for their thoughtful review!

---

### Meta-Review · Area_Chair_pCJB · 2023-09-19

**Recommendation:** 5

**Metareview:**

The paper is clear, well written, and presents significant results. It combines sound modeling, thorough linguistic understanding, and meaningful connections to human cognition. It is particularly well suited to represent this track.

---

### Decision · Program_Chairs · 2023-10-07

**Decision:**

Accept-Main

**Comment:**

The paper is clear, well written, and presents significant results. It combines sound modeling, thorough linguistic understanding, and meaningful connections to human cognition. It is particularly well suited to represent this track.